# BplMYB46 from *Betula platyphylla* Can Form Homodimers and Heterodimers and Is Involved in Salt and Osmotic Stresses

**DOI:** 10.3390/ijms20051171

**Published:** 2019-03-07

**Authors:** Yan-Min Wang, Chao Wang, Hui-Yan Guo, Yu-Cheng Wang

**Affiliations:** 1State Key Laboratory of Tree Genetics and Breeding, Northeast Forestry University, Harbin 150040, China; wangyanmin1919@163.com (Y.-M.W.); wzyrgm@163.com (C.W.); 2Key Laboratory of Fast-Growing Tree Cultivating of Heilongjiang Province, Forestry Science Research Institute of Heilongjiang Province, Harbin 150040, China; 3College of Forestry, Shenyang Agricultural University, Shenyang 110866, China

**Keywords:** *Betula platyphylla*, MYB, dimerization, yeast two-hybrid, stress response

## Abstract

MYB proteins play important roles in the regulation of plant growth, development, and stress responses. Overexpression of *BplMYB46* from *Betula platyphylla* improved plant salt and osmotic tolerances. In the present study, the interaction of eight avian myeloblastosis viral oncogene homolog (MYB) transcription factors with BplMYB46 was investigated using the yeast two-hybrid system, which showed that BplMYB46 could form homodimers and heterodimers with BplMYB6, BplMYB8, BplMYB11, BplMYB12, and BplMYB13. Relative beta-glucuronidase activity and chromatin immunoprecipitation assays showed that the interaction between BplMYB46 and the five MYBs increased the binding of BplMYB46 to the MYBCORE motif. A subcellular localization study showed that these MYBs were all located in the nucleus. Real-time fluorescence quantitative PCR results indicated that the expressions of *BplMYB46* and the five *MYB* genes could be induced by salt and osmotic stress, and the *BplMYB46* and *BplMYB13* exhibited the most similar expression patterns. *BplMYB46* and *BplMYB13* co-overexpression in tobacco using transient transformation technology improved tobacco’s tolerance to salt and osmotic stresses compared with overexpressing *BplMYB13* or *BplMYB46* alone. Taken together, these results demonstrated that BplMYB46 could interact with five other MYBs to form heterodimers that activate the transcription of target genes via an enhanced binding ability to the MYBCORE motif to mediate reactive oxygen species scavenging in response to salt and osmotic stresses.

## 1. Introduction

The MYB family is one of the largest transcription factors (TFs) families in plants. MYBs have functions in various biological processes, such as regulating flavonoid biosynthesis, controlling cell differentiation, responding to hormone stimulus, mediating the cell wall biosynthesis, and enhancing or reducing biotic and abiotic stress tolerance of plants. The overexpression of *MYB113* or *MYB114* increased anthocyanin pigment levels in Arabidopsis [1]. AtMYB23 plays a role in the root epidermal cell type patterning in Arabidopsis [2]. An *MYB* gene, *MYBH*, enhanced hypocotyl elongation in Arabidopsis by promoting auxin accumulation [3]. A series of studies demonstrated that *AtMYB46* is a switch gene in the regulation of secondary wall deposition [4,5,6,7,8,9]. Genomic and transcriptomic analyses demonstrated that *GmMYB* (*Glycine max*) is related to resistance to *Phakopsora pachyrhizi* infection [10]. Overexpression of *GmMYBJ1* increased tolerance to drought and cold stresses in soybean [11].

MYB proteins share a conserved MYB DNA-binding domain that binds to cis-acting elements of transcription factor genes and a diverse C-terminal modulator region to regulate the protein’s activity. The DNA-binding domain and modulator region in MYBs are critical for protein–protein interactions, which can jointly regulate the growth and development of plants [12]. DmMYB (*Drosophila melanogaster*) interacts with CREB-binding proteins (CBPs), which promotes efficient progression through mitosis [13]. B-MYB forms a complex with itself to influence its transcriptional activity [14]. A study indicated that the MYB proteins interact with R/B-like bHLH proteins, jointly controlling the phenylpropanoid biosynthetic pathways, epidermal cell differentiation, and cell patterning in root hair or trichome development [15]. AtMYB44 interacts with regulatory components of ABA receptor 1 (RCAR1)/PYR1-like protein 9 (PYL9) as an abscisic acid (ABA) receptor in Arabidopsis, and negatively regulates ABA signaling [16]. The interaction between of MYB75 and KNOTTED ARABIDOPSIS THALIANA7 (KNAT7, a KNOX family protein) can regulate secondary cell wall biosynthesis [17]. In a recent study, bimolecular fluorescence complementation (BiFC) assay showed that MYB5 and MYB14 physically interact and play synergistic roles in inducing proanthocyanidin (PA) accumulation in *Medicago truncatula* [18]. GhJAZ2 (jasmonate zim-domain protein 2) can negatively regulate cotton fiber initiation via interacting with the R2R3-MYB transcription factor GhMYB25-like [19]. However, the interactions among MYB proteins or between MYBs and other proteins have rarely been studied. Such studies are important to illustrate the functions of these proteins in the growth and development of plants.

A previous study found that *BplMYB46*, an MYB gene from *Betula Platyphylla* (northeast white birch in China), enhances tolerance to salt and osmotic stresses when overexpressed in transgenic plants [20]. In the present study, we further studied the functions and interactions of BplMYB46. The yeast two-hybrid system demonstrated that BplMYB46 could form homodimers or heterodimers. Beta-glucuronidase (GUS) activity and chromatin immunoprecipitation (ChIP) assays further demonstrated that BplMYB46 could interact with BplMYB6, BplMYB8, BplMYB11, BplMYB12, and BplMYB13. Quantitative real-time reverse transcription PCR (qRT-PCR) indicated that *BplMYB46* and *BplMYB13* have similar expression patterns under salt and osmotic stresses. Co-overexpressing *BplMYB13* and *BplMYB46* improved the tolerance of transgenic tobacco to stress by enhancing the expression of their target genes. Our study provides insights into the important role of the interaction of BplMYB46 with other proteins in response to various stresses in plants.

## 2. Results and Discussion

### 2.1. Sequence and Phylogenetic Analyses of Eight MYBs

The cDNA sequences of eight new MYB transcription factors were obtained from the birch transcriptome (GenBank accession numbers: MK512591–MK512598). These MYBs encode proteins ranging from 240 to 341 amino acids (aa), with predicted molecular weights of 28.9 to 38.8 kDa and pI values from 5.38 to 8.66 (Appendix A). Domain prediction showed that these eight MYBs do not have transmembrane domains. At the amino acid sequence level, the coding regions of these MYBs shared 41 to 70% sequence identity. Multiple sequence alignments of the eight MYB proteins with BplMYB46 protein from northeast white birch in China were performed to examine the structures of the MYB transcription factors (Appendix A). BplMYB46 shared two conserved N-terminal DNA-binding R regions with the eight MYBs, showing that they belong to the R2R3-MYB family of transcription factors. The evolutionary relationships of eight MYB proteins in *B. platyphylla* and all MYB proteins in *Arabidopsis thaliana* were analyzed based on an NJ phylogenetic tree (Appendix A). The results showed that the eight MYB proteins are close to Arabidopsis MYB proteins related to stress tolerance, xylem formation and pollen development in plants [21,22,23,24,25], suggesting the eight MYB proteins may have similar functions.

### 2.2. Analysis of the Dimerization of BplMYB46

The yeast two-hybrid (Y2H) system is an effective method to study protein–protein interactions [15,26,27]. In the present study, the interactions of BplMYB46 with the eight MYBs were analyzed using Y2H. First, when the eight pGBKT7-MYBs were used as baits and pGADT7-Rec-BplMYB46 was used as the prey, yeast expressing BplMYB6, BplMYB8, BplMYB11, BplMYB12, and BplMYB13 could grow on QDO/X-α-Gal medium (Figure 1A), indicating that these five MYBs might interact with BplMYB46. When pGBKT7-BplMYB46-N (lacking the activation region) was used as the bait and the eight pGBKT7-MYBs were used as preys, BplMYB46, BplMYB6, BplMYB8, BplMYB11, BplMYB12 and BplMYB13 could grow on QDO/X-α-Gal medium (Figure 1B), further illustrating that BplMYB46 can form homodimers with itself and heterodimers with the five MYBs.

### 2.3. Verification by Transient Expression Assays

To verify the above interactions identified by Y2H, BplMYB6, BplMYB8, BplMYB11, BplMYB12, and BplMYB13 were used for further analysis. The pROK2-*BplMYB* construct was used as an effector, and the fused MYBCORE (CAGTTA)-minimal 35S promoter-GUS was used as the reporter (Figure 2A). Equal ratios of pROK2-*BplMYB46* and the five other pROK2-*BplMYB* were used as double-effectors, respectively. To eliminate the effect of activation by BplMYB46-BplMYB46 homodimers and the combination of BplMYB46 with one of the MYBs, which do not heterodimerize in yeast, the pROK2-*BplMYB46* effector and pROK2-*BplMYB46*:pROK2-*BplMYB7*, as a double-effector, were used as controls, respectively. The effector and a reporter were co-transformed into tobacco leaves using the particle bombardment method (Figure 2B). The relative GUS activity of interaction of BplMYB46 and BplMYB6, BplMYB8, BplMYB11, BplMYB12, and BplMYB13 via the MYBCORE motif was significantly higher than that of only BplMYB46. Among them, the relative GUS activity of the interaction between BplMYB46 and BplMYB13 to MYBCORE was the highest. However, the relative GUS activity of BplMYB46:BplMYB7 bound to MYBCORE was similar to that of BplMYB46 only. Some studies indicated that a high level of relative GUS activity reflected a strong binding ability of proteins and specific target motifs [26,27,28]. Thus, our GUS results suggested that the combination of BplMYB46 and the five MYBs could enhance the binding ability to the MYBCORE motif.

### 2.4. ChIP Analysis 

ChIP analysis was used to determine the binding ability of BplMYB46 and the five MYBs to the MYBCORE motif (Figure 3). The binding abilities of the interactions of the BplMYB46-GFP protein and other five MYBs-GFP proteins to the MYBCORE motif were analyzed using real-time PCR. The results indicated that the enrichment of the MYBCORE motif under the interaction of BplMYB46 with the other MYBs was approximately 2.5–4 fold higher than the enrichment achieved from the *MYB46* promoter. The enrichment of the MYBCORE motif in response to the BplMYB46 protein alone was approximately 1.5 folds higher than the enrichment of the *BplMYB46* promoter. Thus, the binding abilities of BplMYB46-GFP when interacting with the other five MYBs-GFP proteins to the MYBCORE motif were stronger than the binding between BplMYB46-GFP and the MYBCORE motif. Our results demonstrated that the interaction between BplMYB46 and the other five MYBs enhanced their binding to the MYBCORE motif. 

### 2.5. Subcellular Localization of BplMYB46, 6, 8, 11, 12, and 13

To study the subcellular localization of the MYBs, the *MYB*-*GFP* fusion genes and the *GFP* gene were separately transformed into onion epidermal cells using the particle bombardment method. GFP alone was distributed throughout the transformed cells, whereas the six MYB-GFP fusion proteins were exclusively localized to the nucleus (Figure 4), which suggested that BplMYB46 and the other five MYBs are nuclear proteins. 

### 2.6. Expression Patterns of BplMYB46, 6, 8, 11, 12, and 13 in Response to Abiotic Stresses 

To investigate the expression patterns of *BplMYB46*, *6*, *8*, *11*, *12*, and *13* in response to salt and osmotic treatments, a qRT-PCR analysis was performed (Figure 5). Under salt stress, the expression levels of *BplMYB46*, *6*, *8*, *11*, *12*, and *13* gradually increased at 6, 12, and 24 h, reaching a peak level at 24 h and then rapidly decreased at 48 h. Interestingly, the expression patterns of *BplMYB46, 12*, and 13 were similar, and their expression levels were higher than those of *BplMYB6*, *8*, and *11* at 24 h (Figure 5A). Under osmotic stress, the expression analysis showed that the expression patterns of these genes were clustered into two types (Figure 5B). In the first type, the expression levels of *BplMYB46*, *6*, *8*, *12*, and *13* increased gradually at 6 and 12 h, reaching a peak level at 12 h; thereafter, their levels decreased. Interestingly, the expression pattern of *BplMYB46* was more similar to that of *BplMYB13* than to that of *BplMYB6*, *8*, and *12*. In the second type, the expression of *BplMYB11* gradually increased at 6, 12, and 24 h, reaching a peak level at 24 h, and then rapidly decreasing at 48 h. These results showed that the expressions of *BplMYB46* and the 5 *MYB* genes could be induced by salt or osmotic stress. In addition, we found that *BplMYB46* and *BplMYB13* shared very similar expression patterns under salt or osmotic stress. 

### 2.7. Plants Overexpressing BplMYB46 and BplMYB13 Display Alleviated Oxidative Stress and Diminished Cell Membrane Damage 

Our study showed that BplMYB46 could interact with BplMYB6, 8, 11, 12, and 13, and the expression pattern of *BplMYB46* was very similar to that of *BplMYB13* under salt or osmotic stress. Therefore, BplMYB13 was selected for further study. We tested whether co-overexpression of *BplMYB46* and *BplMYB13* could improve abiotic stress tolerance compared with overexpression of *BplMYB13* or *BplMYB46*, separately. In this experiment, the leaves of *BplMYB46*–*BplMYB13* co-overexpressing plants, *BplMYB13* overexpressing plants, *BplMYB46* overexpressing plants, and WT (wild-type) tobacco plants were stained using nitroblue tetrazolium (NBT), 3,3′-diaminobenzidine (DAB), and Evans blue under normal conditions, and under salt and osmotic stresses. NBT and DAB staining reveal the cellular levels of O_2_^−^ and H_2_O_2_, two of the main ROS species involved in stress signaling and oxidative injuries [29]. Under control conditions, no obvious difference in NBT and DAB staining was observed among *BplMYB46* and *BplMYB13* co-overexpressing plants, *BplMYB13* or *BplMYB46* only overexpressing plants, and WT tobacco. By contrast, under NaCl or mannitol treatments at 6 h, the leaves of *BplMYB46* and *BplMYB13* co-overexpressing plants exhibited less blue and brown staining compared with those from plants overexpressing *BplMYB13* or *BplMYB46* alone and WT plants (Figure 6A,B). Evans blue staining can reflect cell death in plants, as manifested by damage to the cell membrane [30]. The results showed that the *BplMYB46* and *BplMYB13* co-overexpressing transgenic plants displayed less intense blue staining compared with plants overexpressing *BplMYB13* or *BplMYB46* alone or WT plants under salt and osmotic stress conditions (Figure 6C). These results showed that *BplMYB46* and *BplMYB13* co-overexpressing transgenic plants had enhanced abilities to scavenge ROS and had decreased levels of cell death, demonstrating that the interaction between BplMYB46 and BplMYB13 could increase plant stress resistance.

### 2.8. Physiological Characterization of BplMYB46 and BplMYB13 Co-Overexpressing Plants 

Superoxide dismutase (SOD), peroxidase (POD), and glutathione-S-transferase (GST) activities, H_2_O_2_ and malondialdehyde (MDA) contents, and electrolyte leakage have been used to analyze stress tolerance associated with TFs in plants [31,32]. In this study, the activities of SOD, POD, and GST, the H_2_O_2_ and MDA contents, and electrolyte leakage were used to detect the resistance of plants co-overexpressing *BplMYB46* and *BplMYB13* to salt and osmotic stresses; water was used the control (Figure 7). Under control conditions, the activities of SOD, POD, and GST in *BplMYB46* and *BplMYB13* co-overexpressing, *BplMYB13* or *BplMYB46* only overexpressing plants, and WT plants showed almost no differences. However, under salt and osmotic stresses at 6 h, the activities of SOD, POD, and GST in the *BplMYB46* and *BplMYB13* co-overexpressing plants were significantly higher compared with those in *BplMYB13* or *BplMYB46* only overexpressing plants, and WT plants (Figure 7A–C). Under control conditions, there was hardly any difference in the H_2_O_2_ and MDA contents between the *BplMYB46* and *BplMYB13* co-overexpressing plants, the *BplMYB13* or *BplMYB46* only overexpressing plants, and the WT plants. However, under salt and osmotic stresses at 6 h, in the *BplMYB46* and *BplMYB13* co-overexpressing plants, the H_2_O_2_ and MDA contents were significantly lower than those in the *BplMYB13* or *BplMYB46* only overexpressing plants and WT plants (Figure 7D,E). The H_2_O_2_ and MDA contents (levels of O_2_^−^) can negatively reflect the ROS scavenging ability of plants; therefore, our results demonstrated that the interaction between *BplMYB46* and *BplMYB13* could enhance the plants’ ROS scavenging ability. Electrolyte leakage has been used as a measure of cell death. Under control conditions, there was no significant difference among the *BplMYB46* and *BplMYB13* co-overexpressing plants, the *BplMYB46* overexpressing plants, and WT plants. However, under salt and osmotic stresses at 6 h, the *BplMYB46* and *BplMYB13* co-overexpressing transgenic plants had significantly lower levels of electrolyte leakage than those in the *BplMYB46* overexpressing plants and WT plants, indicating that the combination with *BplMYB13* could decrease cell death (Figure 7F). These results suggested that the interaction of BplMYB46 with BplMYB13 could enhance tolerance to salt or osmotic stresses by enhancing the ROS scavenging ability and decreasing cell death in plants. 

### 2.9. The Relative Expression of Target Genes 

To further analyze whether the interaction of BplMYB46 with BplMYB13 could affect the expression of target genes, the relative expression levels of *SOD*, *POD*, and *GST* in *BplMYB46* and *BplMYB13* co-overexpressing plants, *BplMYB46* or *BplMYB13* only overexpressing plants, and WT plants under salt and osmotic stresses were examined. Under salt or osmotic treatment, the relative expression levels of *SOD*, *POD*, and *GST* genes were significantly higher in the *BplMYB46* and *BplMYB13* co-overexpressing plants than in the *BplMYB46* or *BplMYB13* only overexpressing plants and WT plants compared with those under fresh water treatment (Figure 8). Analysis of the DNA sequences on the PLACE website showed that the MYBCORE motif exists in the promoters of the five genes. Thus, our results suggested that the interaction of BplMYB46 with BplMYB13 might enhance the expression levels of *SOD*, *POD*, and *GST* genes by binding to the MYBCORE cis-acting element.

## 3. Materials and Methods 

### 3.1. Plant Materials and Growth Conditions

Tobacco seedlings were planted on half Murashige–Skoog (1/2 MS) medium in a growth chamber with a 16/8 h light/dark cycle and an average temperature of 25 °C.

Northeast white birch seeds were planted in pots containing a mixture of perlite/soil (2:1) in a greenhouse under controlled conditions of 16/8 h light/dark, an average temperature of 25 °C, and 70–75% relative humidity. 

### 3.2. Sequence Analysis of MYB Transcription Factors

The cDNA sequences of eight MYB transcription factors were obtained from the birch transcriptome. The theoretical molecular weight (MW) and isoelectronic point (pI) predictions for each deduced MYB were calculated using the ExPASy compute pI/Mw tool (http://www.expasy.org/tools/protparam.html). Transmembrane domains were predicted using the TMHMM server 2.0 (http://www.cbs.dtu.dk/services/TMHMM/). Multiple sequence alignments of BplMYB46 with the eight MYBs were performed using the ClustalW program (http://www.ebi.ac.uk/clustalw/). An unrooted phylogenetic tree was constructed using MEGA5.1, according to the neighbor-joining (NJ) method with 1000 bootstrap replicates. The sequences of the Arabidopsis MYB domain proteins were downloaded from the Arabidopsis genome TAIR 9.0 (http://www.arabidopsis.org/).

### 3.3. Heterodimer and Homodimer Assays for BplMYB46

The open reading frame (ORF) of the *BplMYB46* cDNA, without the terminal codon, was inserted into the pGADT7-Rec vector via its SmaI (Promega, Madison, WI, USA) site as the prey construct. The eight *BplMYBs*, without their terminal codons, were inserted into pGBKT7 and used as bait constructs. The pGBKT7 empty vector and pGADT7-Rec-BplMYB46, and pGBKT7 and pGADT7-Rec empty vectors were used as negative controls. Baits and preys were transformed into the yeast strain Y2HGold, and then plated on SD/-Trp/-Leu (double dropout (DDO)) and SD/-Trp/-Leu /-His/-Ade/X-α-Gal (quadruple dropout (QDO) media at 30 °C for 3–5 days. 

A truncated fragment of the BplMYB46 cDNA, lacking the region encoding the transactivation region, was constructed into the pGBKT7 vector as the bait, and BplMYB46 and the eight MYBs lacking their terminal codons were constructed into the pGADT7-Rec as preys. Construct pGBKT7-BplMYB46 and pGADT7-Rec empty vector, and pGBKT7 and pGADT7-Rec empty vectors were used negative controls. Vector pGBKT7-53 and pGADT7-Rec-T were used positive controls. Vector pGBKT7-53 encodes the Gal4 DNA-BD fused with murine p53; pGADT7-T encodes the Gal4 AD fused with SV40 large T-antigen. The p53 and large T-antigen are known to interact in a yeast two-hybrid assay; therefore, mating Y2HGold (pGBKT7-53) with Y187 (pGADT7-T) allows the cells to grow in QDO mediums/X-α-Gal. Baits and preys were transformed into Y2H cells, and were grown on DDO and QDO/X-α-Gal media at 30 °C for 3–5 days. All the primer sequences used are shown in Appendix A. 

### 3.4. Transient Expression Assays

The full length ORFs of *BplMYB6*, *BplMYB7*, *BplMYB8*, *BplMYB11*, *BplMYB12*, and *BplMYB13* were inserted separately into vector pROK2. An equal proportion of pROK2-*BplMYB46* and other pROK2-*BplMYBs* formed the double-effectors, respectively. The fused MYBCORE-minimal 35S CaMV promoter-GUS construct was used as the reporter. The effector and reporter vector were co-transformed into tobacco leaves using the particle bombardment method (Bio-Rad, Hercules, CA, USA). The effector of pROK2, pROK2-*BplMYB46* and pROK2-*BplMYB46*: pROK2-*BplMYB7* served as controls, respectively. The firefly luciferase gene driven by the CaMV 35S promoter was also co-transformed as an internal control for normalization of the transformation efficiency. The GUS activity from the empty pROK2 vector as an effector was set to 1. GUS and luciferase activities were determined according to previously published methods [33], and the data are shown as the averages of three biological replicates. The primer sequences are shown in Appendix A.

### 3.5. Chromatin Immunoprecipitation (ChIP) Analysis 

The ORFs of *BplMYB6*, *BplMYB8*, *BplMYB11*, *BplMYB12*, *BplMYB13*, and *BplMYB46* without their terminal codons were inserted separately into vector pBI121, upstream of GFP, and the six pBI121-*MYB*-*GFPs* were transformed separately into EHA 105 Agrobacterium competent cells. pBI121-*BplMYB* and pBI121-*BplMYB46* were co-transformed into birch by transient Agrobacterium-mediated transformation [34]. At 48 h after transformation, the transgenic plants were collected for the ChIP assay. DNA and protein were cross-linked using 1% formaldehyde, then the chromatin was sheared into 200–750 bp fragments using sonication, and 10% of the sample was saved as the input control. The remaining sonicated chromatin was divided into two parts, which were respectively added with anti-GFP antibodies (ChIP+) and no antibody (negative control, ChIP−). The DNA truncated fragments were released by reversing the cross-linking at 65 °C for 6 h, and were then digested by proteinase K (0.2 mg/mL) for 1 h at 45 °C to remove any residual proteins. The immunoprecipitated DNA was purified using chloroform extraction. The enrichment of the truncated promoter, including the MYBCORE motif was determined on the chromatin DNA samples after immunoprecipitation, using real-time PCR. The values for the bound MYBCORE motif were normalized against those of the truncated *BplMYB46* promoter. Three biological replications were used in this experiment. The primer sequences are shown in Appendix A.

### 3.6. Subcellular Localization Analysis

The pBI121-BplMYBs-GFP fusion gene and pBI121-GFP (control) were transiently expressed in onion epidermal cells using the particle bombardment (Bio-Rad) method. The transformed cells were then cultured on MS medium for 24–48 h and analyzed using a confocal laser-scanning microscope at 488 nm (LSM410, Zeiss, Jena, Germany).

### 3.7. Plant Stress Treatments and Real-Time Reverse Transcription (RT)-PCR 

Two-month-old birch seedlings were collected at the same time after treatment for 6, 12, 24, and 48 h under 200 mM NaCl (salt stress) or 200 mM Mannitol. Mannitol is a naturally occurring sugar alcohol that can cause osmotic stress and has been widely used for studies of osmotic stress in plants. Seedlings watered with fresh water were used as controls. Total RNA was isolated from birch using the CTAB method and treated with DNase I (Takara Bio Inc., Shiga, Japan) and RNase free to remove DNA contamination. The total RNA was treated with DNase I and was reverse transcribed into cDNA using a PrimeScript™ RT reagent Kit (Takara Bio Inc.) and real-time PCR was performed for *BplMYB46*, *BplMYB6*, *BplMYB8*, *BplMYB11*, *BplMYB12*, and *BplMYB13*. The sequences of the primers used are listed in Appendix A. Real-time quantitative RT-PCR was performed with the SYBR Premix Ex Taq™ kit. DNA were amplified, according to the following procedure: 94 °C for 30 s; 45 cycles at 94 °C for 12 s, 58 °C for 30 s, and 72 °C for 45 s; followed by 79 °C, 1 s for plate reading. A melting curve was generated for each sample to assess the purity of the amplified products. The average values of the cycle thresholds (Ct) of the α-tubulin and ubiquitin genes were used as internal references. The relative expression ratios were calculated from the Ct values according to the delta-delta Ct method [35]. The relative transcription level was calculated as the transcription level under stress treatment divided by the transcription level under control conditions. Three independent biological replicates were performed in this experiment. 

### 3.8. Stress Tolerance Analysis of the Interaction of BplMYB46 with BplMYB13

The overexpression constructs pROK2-*BplMYB46* and pROK2-*BplMYB13* were co-transformed, and pROK2-*BplMYB13* and pROK2-*BplMYB46* were separately transformed into tobacco using transient *Agrobacterium*-mediated transformation. The tobacco co-overexpressing *BplMYB46* and *BplMYB13*, overexpressing *BplMYB13* or *BplMYB46* only, and the WT were treated with 200 mmol/L NaCl or 200 mmol/L Mannitol stresses for 6 h; treatment with water only was used as a control. The detached leaves from tobacco were infiltrated with NBT (1.0 mg/mL) and DAB (1.0 mg/mL) following published procedures [29], and Evans blue (1.0 mg/mL) staining was performed to detect cell death following a previously published protocol [36]. SOD, POD, GST, H_2_O_2_, MDA, and electrolyte leakage measurements were conducted according to previously published methods [27,31]. Three biological replicates were performed.

### 3.9. Expression Analysis of Target Genes 

Tobacco co-overexpressing *BplMYB46* and *BplMYB13*, tobacco overexpressing *BplMYB13* or *BplMYB46* only, and WT tobacco were collected after 6 h of treatment with 200 mM NaCl or 200 mM Mannitol, and seedlings watered with fresh water were used as controls. Three independent biological replicates were performed. Total RNA was extracted and treated with DNase I before being reverse transcribed into cDNA using a PrimeScript™ RT reagent Kit (Takara). Real-time PCR was then performed for two *SOD*, two *POD*, and 1 *GST* (GenBank accession number: MK532397) genes. The sequences of the primers used are listed in Appendix A.

### 3.10. Statistical Analysis

Analysis of variance (ANOVA) was used to analyze the relative GUS activity. All statistical analyses were performed using SPSS software (IBM Corp., Armonk, NY, USA), version 18.0. 

## 4. Conclusions

Previously, we reported that BplMYB46 is involved in abiotic stress tolerance and secondary wall deposition. In the present study, we further showed that BplMYB46 could homodimerize with itself and heterodimerize with the BplMYB6, BplMYB8, BplMYB11, BplMYB12, and BplMYB13. The interaction of BplMYB46 and the five MYB proteins could improve the binding ability of BplMYB46 to the MYBCORE motif. In addition, *BplMYB6*, *BplMYB8*, *BplMYB11*, *BplMYB12*, and *BplMYB13* exhibited similar expression patterns to that of *BplMYB46* in birch under salt and osmotic treatment. A subcellular localization study showed that these MYBs were all targeted to the nucleus. *BplMYB46* and *BplMYB13* co-overexpression in tobacco improved the plants’ tolerance to salt and osmotic stresses compared with *BplMYB46* overexpression alone and further demonstrated that the interaction of BplMYB46 with BplMYB13 might increase plant stress tolerance.

## Figures and Tables

**Figure 1 ijms-20-01171-f001:**
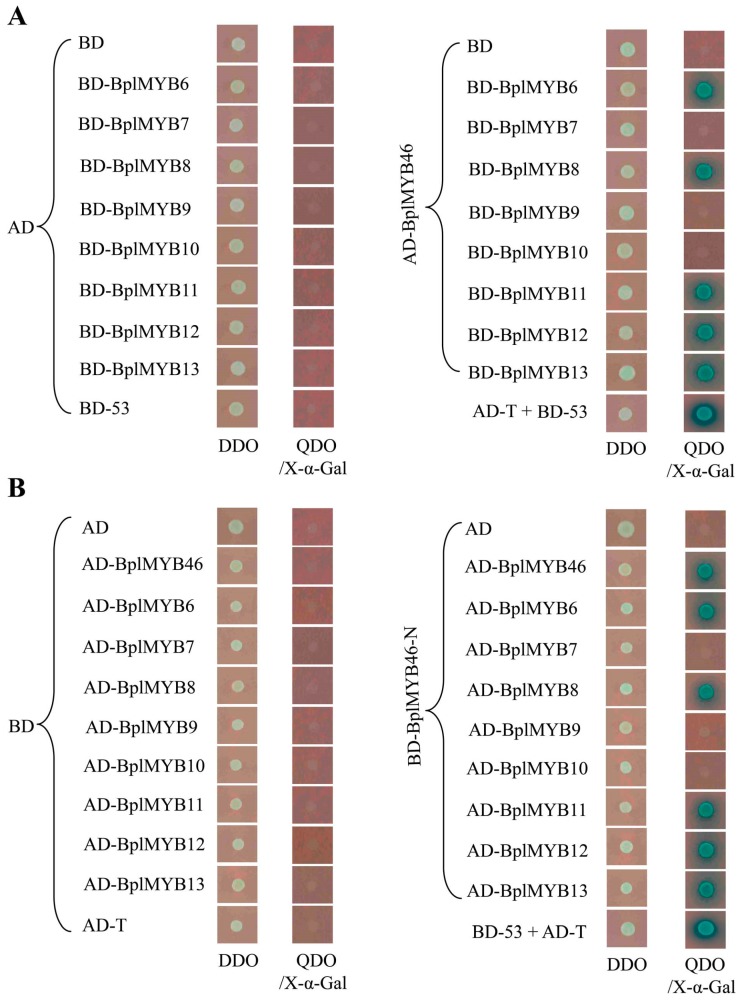
Homodimeric and heterodimeric analysis of BplMYBs using a yeast two-hybrid assay (Y2H). (**A**) BplMYB46 was cloned into a pGADT7-Rec vector (AD-prey) and interacted separately with the eight MYB proteins, which were fused to the GAL4 DNA binding domain in the yeast pGBKT7 vector (BD-baits). Empty AD-prey and each BD-bait were used as negative controls; AD-BplMYB46-prey and empty BD-bait were used as negative controls; and AD-T-prey and BD-53-bait were used as positive controls. (**B**) BplMYB46-N (without its activation regions) was cloned into pGBKT7 vector (bait) and interacted with itself and with the other eight MYB proteins cloned into the pGADT7-Rec vector (preys). Empty BD-bait and each AD-prey were used as negative controls; BD-BplMYB46-N-bait and empty AD-prey were used as negative controls; BD-53-bait and AD-T-prey were used as positive controls. The yeast cells were grown on SD/-Trp/-Leu (double dropout (DDO)) and selective dropout media: SD/-Trp/-Leu/-His/-Ade/X-α-Gal (quadruple dropout (QDO)/X-α-Gal).

**Figure 2 ijms-20-01171-f002:**
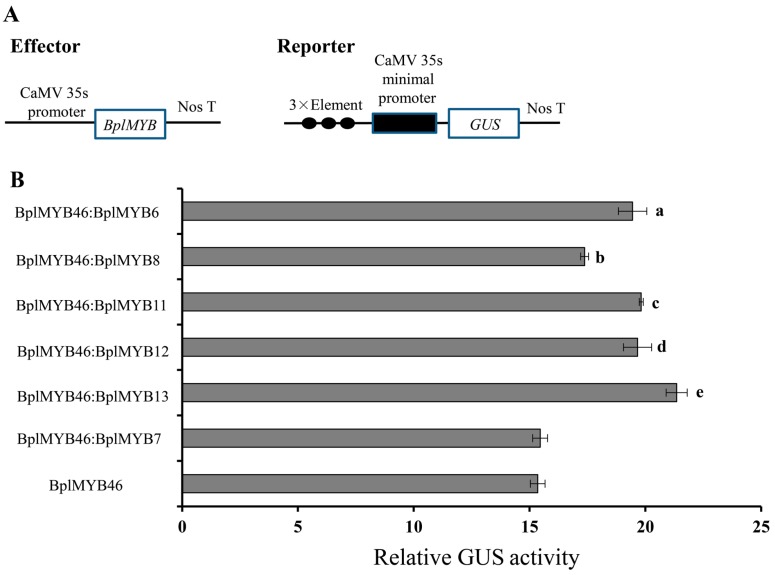
Beta-glucuronidase (GUS) activity analysis of the binding of BplMYB46 and five other MYBs to the MYBCORE sequence. (**A**) Schematic diagram of the effector and reporter constructs used in the GUS analysis. Triple tandem copies of the MYBCORE were fused with the 35S CaMV-46 minimal promoter and cloned into pCAMBIA1301 to drive expression of the *GUS* gene as the reporter construct. The coding sequences of *BplMYBs* were cloned into pROK2 under the control of the 35S promoter as the effector constructs. (**B**) The GUS activity assay results. Each effector and the reporter constructs were co-transformed into tobacco leaves. The combination of BplMYB46 with BplMYB7, which do not heterodimerize in yeast, was used as a control. The empty pROK2 vector was used as a negative control. The 35S-luciferase construct was transformed together with the reporter and effector into leaves to normalize the transformation efficiency. The GUS activity resulting from using the empty pROK2 vector as the effector was set to 1. The error bars are the standard deviations, which were calculated from three independent biological repeats. Lower case letters indicate *p* < 0.05.

**Figure 3 ijms-20-01171-f003:**
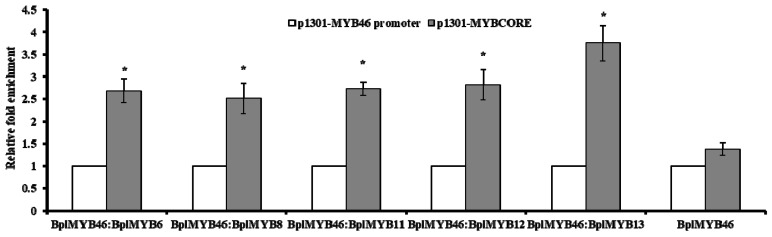
Chromatin immunoprecipitation (ChIP) analysis of BplMYB46 and five other MYBs binding to the MYBCORE motif. Quantitative real-time PCR analysis showing the enrichment of the promoter sequences, including the MYBCORE motif after chromatin immunoprecipitation. The values for the enrichment of the *BplMYB46* promoter sequences were set to 1. Chromatin from whole seedlings was isolated from pBI121-*BplMYB46*-*GFP* and pBI121-*MYB-GFP* birch plants produced by *Agrobacterium tumefaciens*-mediated transient transformation. The error bars indicate the standard deviation (SD) from three biological replicates. * indicates a significant difference (*p* < 0.05).

**Figure 4 ijms-20-01171-f004:**
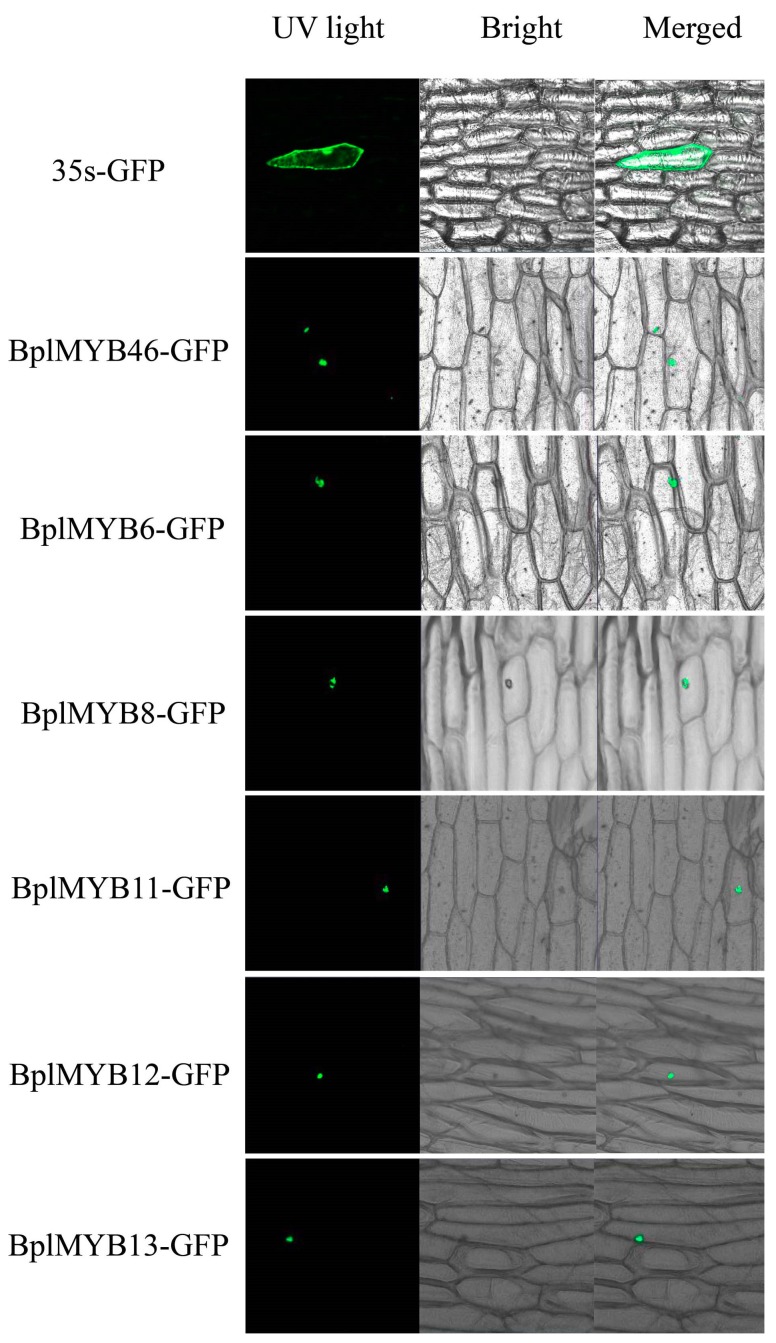
Nuclear localization of BplMYB proteins. The *BplMYB*-*GFP* fusion genes and *GFP* (control) were transiently expressed in onion epidermal cells using the particle bombardment method. The transformed cells were cultured on Murashige–Skoog (MS) medium for 24–48 h and visualized using a confocal microscope at 488 nm.

**Figure 5 ijms-20-01171-f005:**
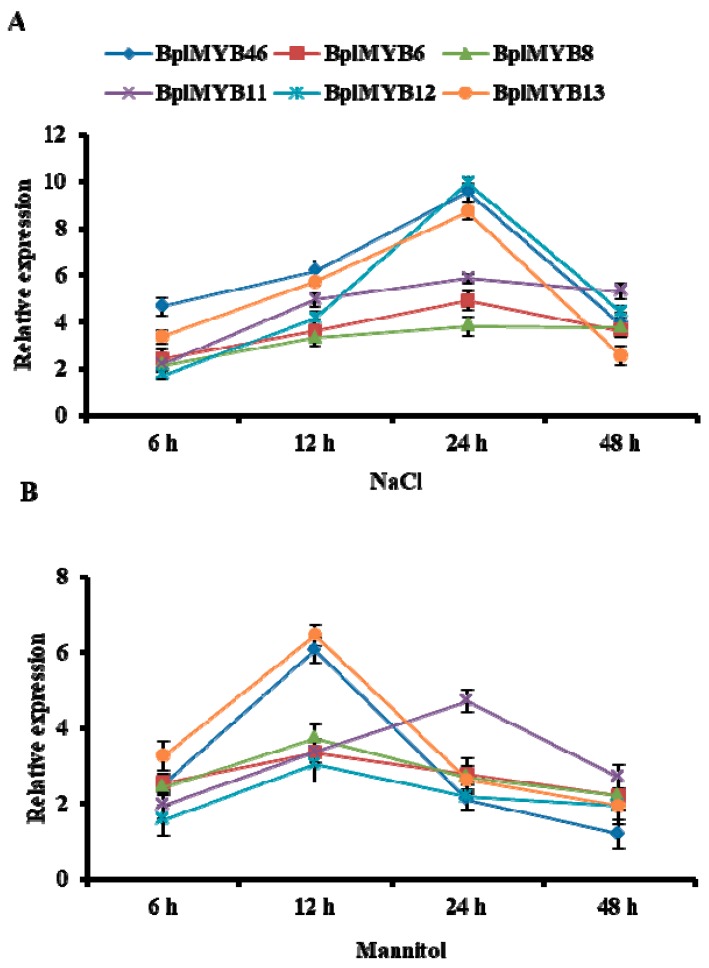
Expression profiles of *BplMYB46*, *BplMYB6*, *BplMYB8*, *BplMYB11, BplMYB12,* and *BplMYB13* under different abiotic stresses. Two-month-old birch seedlings were treated with 200 mM NaCl (**A**) or 200 mM Mannitol (**B**) for different times. Fresh-watered plants were grown in parallel as controls. After these treatments, the stems of the seedlings from each sample were harvested and pooled for real-time reverse transcription (RT)-PCR analyses. The error bars indicate the standard deviation (SD) from three biological replicates.

**Figure 6 ijms-20-01171-f006:**
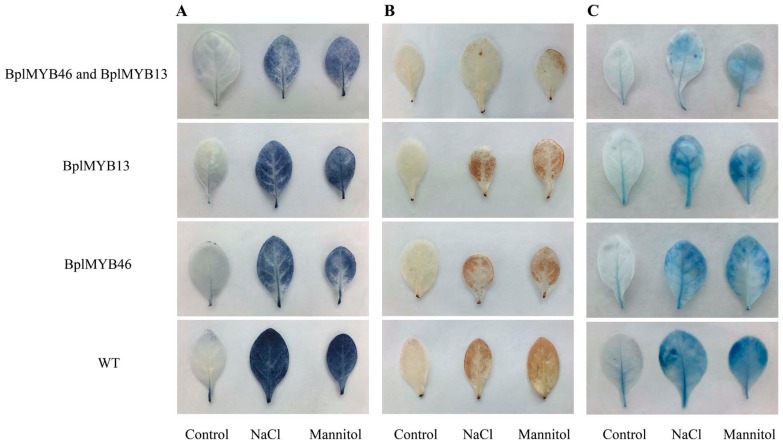
The reactive oxygen species (ROS) levels and cell death in *BplMYB46* and *BplMYB13* transgenic and wild-type tobacco under NaCl and Mannitol treatment. (**A**, **B**) Cellular levels of O_2_^−^ and H_2_O_2_ in *BplMYB46* or *BplMYB13*-transformed and wild-type (WT) plants under normal conditions, and under salt or drought stress. Leaves from *BplMYB13*-transformed and WT plants were pretreated with of NaCl or mannitol, and the levels of O_2_^−^ and H_2_O_2_ were visualized using nitroblue tetrazolium (NBT) and 3,3′-diaminobenzidine (DAB) staining, respectively. (**C**) Evans Blue staining analysis of cell death. The leaves sampled from two-month-old transgenic and WT plants were treated with NaCl or mannitol at 6 h and used for histochemical staining, treatment with water only was used as a control. Each experiment was repeated at least three times, and approximately 10 leaves harvested from multiple seedlings were inspected.

**Figure 7 ijms-20-01171-f007:**
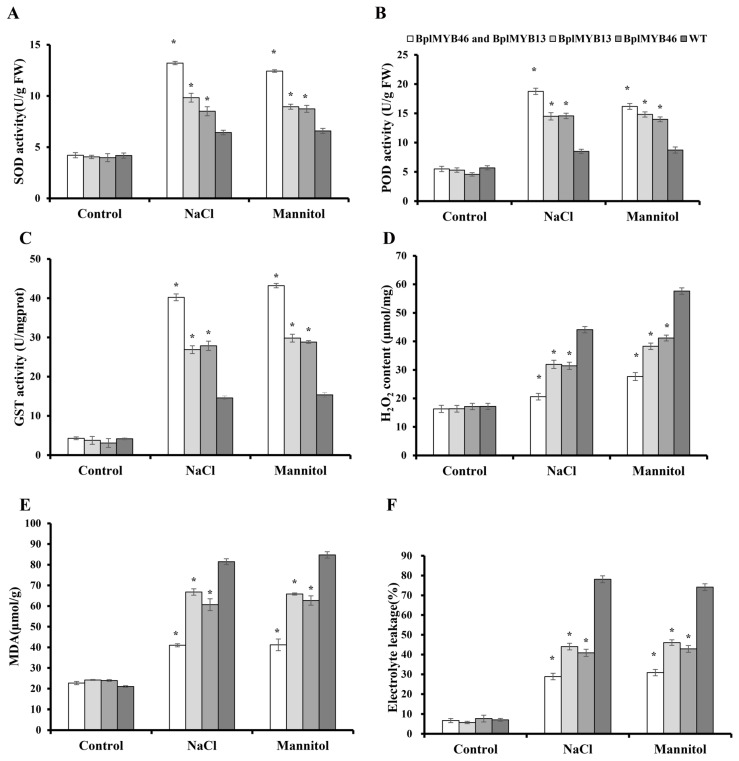
Salt and osmotic stress tolerance analyses of *BplMYB46* and *BplMYB13* transgenic and wild-type (WT) tobacco. Superoxide dismutase (SOD) activity (**A**), peroxidase (POD) activity (**B**), glutathione-S-transferase (GST) activity (**C**), and H_2_O_2_ contents (**D**), malondialdehyde (MDA) contents (**E**), and electrolyte leakage (**F**) of transgenic and WT tobacco under salt and osmotic stress. The seedlings sampled from the transgenic tobacco using transient *Agrobacterium*-mediated transformation and WT plants were treated with NaCl and mannitol at 6 h, treatment with water used as a control. Each experiment was repeated at least three times, and approximately 10 leaves harvested from multiple seedlings were inspected. The error bars indicate the standard deviation (SD) from three biological replicates. * indicates a significant difference (*p* < 0.05).

**Figure 8 ijms-20-01171-f008:**
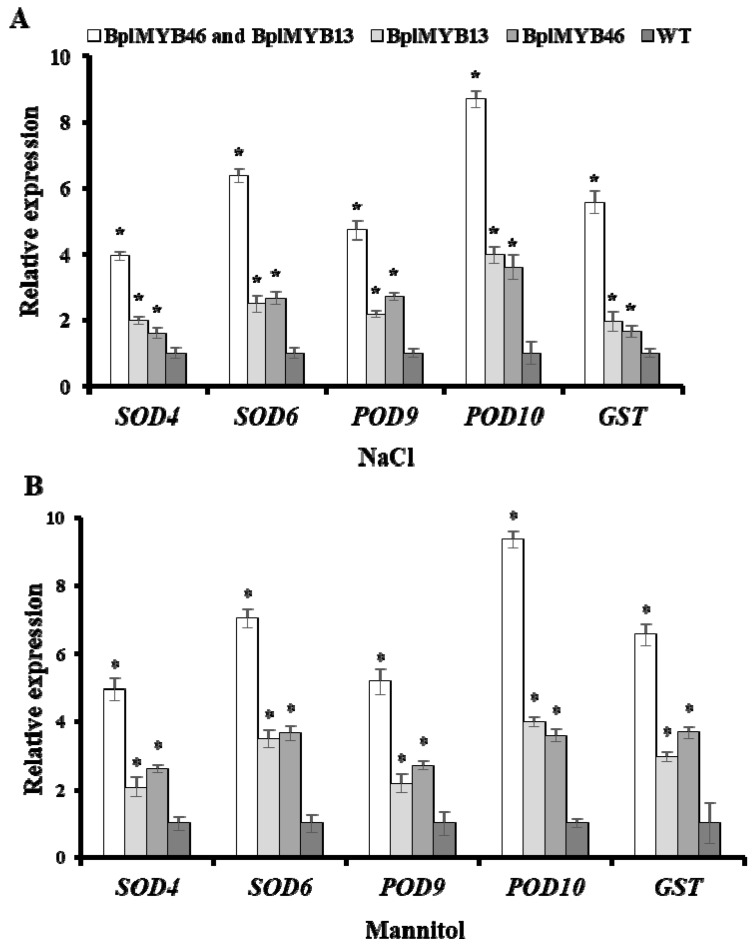
The relative expression analyses of target genes in *BplMYB46* and *BplMYB13* transgenic and wild-type (WT) birch. The relative expression levels of *superoxide dismutase (SOD)*, *peroxidase (POD)* and *glutathione-S-transferase (GST)* genes of transgenic and WT tobacco under salt stress (**A**) under osmotic stress (**B**). The seedlings sampled from the transgenic tobacco using transient *Agrobacterium*-mediated transformation and WT plants were treated with NaCl and mannitol at 6 h, treatment with water only was used as a control. The error bars indicate the standard deviation (SD) from three biological replicates. * indicates a significant difference (*p* < 0.05).

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
