# Peer review of "BplMYB46 from Betula platyphylla Can Form Homodimers and Heterodimers and Is Involved in Salt and Osmotic Stresses"

_ijms, 2019, doi:10.3390/ijms20051171_

Round 1

Reviewer 1 Report

Specificity of gene expression is govern transcription factors (TF) that function in a combinatorial fashion. In angiosperm a the largest subfamily of TF are R2R3-MYBs. Expression of genes encoding the lignin biosynthetic pathway is tightly controlled and the key role in this process is played by members of R2R3-MYBb TFs. In Arabidopsis AtMYB46 and AtMYB83 was shown to govern extensive network of other TFs, such as AtMYB58, AtMYB63, AtMYB103 that directly activate expression of lignin biosynthesis genes or (AtMYB4, AtMYB7 and AtMYB3) or inhibit the expression of negative regulators of biosynthesis. In poplar as many as 23 R2R3-MYB genes are expressed at the highest level during xylem differentiation and functional orthologs of Arabidopsis master switches were identified. In birch (Betula platyphylla Suk.) BplMYB46 was found to participate in tolerance to environmental stresses and secondary wall biosynthesis. Despite that R2R3-MYB TFs played a prominent roles in the regulation of secondary cell wall synthesis little is known about the mechanism of their interactions with their DNA targets and possible cross-interaction within this family as well as with other TF families.

The manuscript “BplMYB46 from Betula platyphylla can form homodimers and heterodimers and is involved in salt and osmotic stresses” by Wang and colleges attempts to face this topic. The authors investigates possible interactions between different members of R2R3-MYB family with BplMYB46, which is important for stress response. My first comment is that the criteria for the selection of genes for analysis have not been clarified. Are there all R2R3-MYBs from birch genome? According to my knowledge, except of BpMYB106 there is no experimental data about other birch MYBs. Author mentioned that cDNA sequences of analyzed MYBs were obtained from birch 80 transcriptomes (in lines 80-81 there is no citation or information about these transcriptomes). I suggest also to analyze homology of eight newly identified R2R3-MYBs to the MYBs from other plant species.

detailed comments Second general remark is that the Figures are inacceptable in a present form. Each Figure should be self-explanatory and contain technical details to enable the readers to understand what is shown without referring to the text. Third, I found that some of applied methods are not suitable I do not support the conclusions (see below). Four, obtained results are not discussed at all.

Particular objections:

1.     Y2H gives high level of false positives; what were the controls? Did the expression of bait and pray was verified by Western blots?

2.     Transient expression i) assays lacks necessary controls

-       negative controls, e.g. combination of BplMYC46 with one of MYBs that do not heterodimerize in yeast

-       protein expression was not confirmed; the level of proteins can be then unequal; as BplMYC46 homodimerize in experiments there could be a different ratio of MYC46-homodimers, MYC46-MYC# heterodimers;

-       the unit “relative GUS activity” is not defined

In consequence the results are not reliable. Since other analyzed BplMYC# can homodimerize GUS expression in all combination may results from activation by MYC46-MYC46 homodimers, and the differences in the level of expression may be a consequence of different level of expression of MYC46 in the presence of the second sequence.

Then the conclusion, such as “interaction between BplMYB46 and BplMYB13 was the strongest” (line 178) is far going.

3. The results of Evans blue staining should be quantified.

Lines 189-190 – “Evans blue staining can reflect the cell death in plants under stress conditions” remove “under stress conditions”

4.  Line 251 “SOD, POD, and GST genes by binding to the MYBCORE cis-acting element” Do promotors of SOD, POD, and GST genes in birch indeed contain MYBCORE element? This regulation can be also indirect. Which GST gene was analyzed?

Detailed comments:

Lines 44-46MYB proteins share a conserved MYB DNA-binding domain that binds to cis-acting elements of transcription factor genes and a diverse C-terminal modulator region to regulate the protein’s activity. These domains in MYBs” – it is not cleat which domains author means

Lines 55-56recent study used a bimolecular fluorescence complementation (BiFC) assay and found” it is not the study that uses the method but the method was used in the study

Line 64 – “enhanced resistance” – it is better to use tolerance not resistance, as the latter applies rather to pathogenesis

Line 72 – “reactive oxygen species (ROS)-related genes” – there is no gene related to ROS

Remove lines 77-79

Lines 84-85 – “Trans-membrane domain prediction showed that these eight MYBs do not have a trans-membrane domain” - transmembrane not trans-membrane; this sentence should be improved;

Line 166 – “The control GFP was distributed” There is no “control GFP”

Line 175-176 – “Plants co-overexpressing BplMYB46 and BplMYB13 display alleviated oxidative and cell membrane damage” change to “In plants overexpressing BplMYB46 and BplMYB13 oxidative stress was alleviated and the cell membrane damage was diminished”

Line 194 – “plants had enhanced abilities to scavenge oxygen” change to “scavenge ROS”

Lines 273-274 – “Baits and preys were transformed into Y2H cells” give the correct yeast strain name

Change “vector pbi121” to vector pBI121

Figure 7 – make legend more visible

Author Response

My first comment is that the criteria for the selection of genes for analysis have not been clarified. Are there all R2R3-MYBs from birch genome? According to my knowledge, except of BpMYB106 there is no experimental data about other birch MYBs. Author mentioned that cDNA sequences of analyzed MYBs were obtained from birch 80 transcriptomes (in lines 80-81 there is no citation or information about these transcriptomes). I suggest also to analyze homology of eight newly identified R2R3-MYBs to the MYBs from other plant species.

Response: Thank you for your suggestions. R2R3-MYBs were found in birch transcriptome, but some nucleotide sequences were repeated, resulting in eight complete cDNA sequences of R2R3-MYBs being obtained for analysis in this study. We uploaded the sequence information of these eight MYB genes to NCBI, obtained the GenBank accession numbers, and added them to our manuscript (Line78). We added the phylogenetic analysis of the eight new MYB protein sequences to all MYBs from Arabidopsis (Line86-90).

Detailed comments Second general remark is that the Figures are inacceptable in a present form. Each Figure should be self-explanatory and contain technical details to enable the readers to understand what is shown without referring to the text.

Response: We have revised the figure legends according to your suggestions.

Particular objections:

1.Y2H gives high level of false positives; what were the controls? Did the expression of bait and pray was verified by Western blots?

Response: We have added the positive control and negative controls in the revision, and the results showed that the positive control grows well in QDO medium and the negative control cannot grow in QDO medium (Figure 1), indicating that the Y2H results are reliable. The interaction of proteins with proteins in QDO indicated that the bait and prey proteins have been expressed. Usually, it is not required to test the expression of bait and prey using western blotting in Y2H according to previous reports and its protocol (Matchmaker® Gold Yeast Two-Hybrid System User Manual).

2.Transient expression i) assays lacks necessary controls

- negative controls, e.g. combination of BplMYC46 with one of MYBs that do not heterodimerize in yeast

Response: Thank you for your suggestions. We added the combination of BplMYB46 with BplMYB7, which do not heterodimerize in yeast, as a control (Figure 2).

- protein expression was not confirmed; the level of proteins can be then unequal; as BplMYC46 homodimerize in experiments there could be a different ratio of MYC46-homodimers, MYC46-MYC# heterodimers;

Response: In the pre-experiments, we tested different ratios of BplMYB46-MYB heterodimers, including the ratios of 1:1, 1:2, 1:3, 2:3, and 3:2, and the results indicated that the equal ratio was best; therefore, the equal ratio of BplMYB46-MYB heterodimers was used in our later experiments. We have added the explanation to our manuscript (Line121, Line337).

- the unit “relative GUS activity” is not defined

Response: We recalculated the relative GUS activity. The empty pROK2 vector as the effector was used as the negative control, and the relative GUS activities were calculated using the GUS activity of each sample divided by the GUS activity of its negative control. Therefore, “relative GUS activity” is a relative ratio that has not unit. We have specified this in the revision (Line145-146, Line343-344). 

In consequence the results are not reliable. Since other analyzed BplMYC# can homodimerize GUS expression in all combination may results from activation by MYC46-MYC46 homodimers, and the differences in the level of expression may be a consequence of different level of expression of MYC46 in the presence of the second sequence.

Response: The interaction between MYB46 and MYB46 was also assessed and used as control to eliminate the effects of activation by MYB46-MYB46 homodimers (Line122-123), which is shown in Figure 2. From Figure 2, it can be see that MYB46-MYB46 could active GUS expression at a significantly lower level than did the other interactions, and this could show that the interaction between MYB46 and other MYBs can form heterodimers to enhance the expression of GUS (Line121-131). 

Then the conclusion, such as “interaction between BplMYB46 and BplMYB13 was the strongest” (line 178) is far going.

Response: We have revised it according to your suggestion (Line208-209).

3. The results of Evans blue staining should be quantified.

Response: Evans blue staining is a qualitative method to investigate cell death, and we have not found a method to quantify the results. However, electrolyte leakage is also used as an index of cell death. Therefore, we added the results of electrolyte leakage analysis to the revised manuscript (Figure 7), the results were consistent with the staining of Evans blue, indicating that the results are reliable.

Lines 189-190 – “Evans blue staining can reflect the cell death in plants under stress conditions” remove “under stress conditions”

Response: We deleted it according to your suggestion.

4.  Line 251 “SOD, POD, and GST genes by binding to the MYBCORE cis-acting element” Do promotors of SOD, POD, and GST genes in birch indeed contain MYBCORE element? This regulation can be also indirect. Which GST gene was analyzed?

Response: Yes, the promoters of SOD, and POD genes contain the MYBCORE element, as reported in the ChIP analysis in our previous study (Guo, H.; Wang, Y.; Wang, L.; Hu, P.; Wang, Y.; Jia, Y.; Zhang, C.; Zhang, Y.; Zhang, Y.; Wang, C.; Yang, C., Expression of the MYB transcription factor gene BplMYB46 affects abiotic stress tolerance and secondary cell wall deposition in Betula platyphylla. Plant biotechnology journal 2017, 15, (1), 107-121). The glutathione transferase (GST) gene was studied in this study. The MYBCORE element on its promoter sequence was analyzed and identified. We uploaded the sequence information of this GST gene to NCBI, obtained the GenBank accession number, and added it to our manuscript (Line404).

Detailed comments:

Lines 44-46 – MYB proteins share a conserved MYB DNA-binding domain that binds to cis-acting elements of transcription factor genes and a diverse C-terminal modulator region to regulate the protein’s activity. These domains in MYBs” – it is not cleat which domains author means

Response: We have revised the text in our manuscript according to your suggestion (Line47).

Lines 55-56 – recent study used a bimolecular fluorescence complementation (BiFC) assay and found” it is not the study that uses the method but the method was used in the study

Response: We have revised this text in our manuscript according to your suggestion (Line57-58).

Line 64 – “enhanced resistance” – it is better to use tolerance not resistance, as the latter applies rather to pathogenesis

Response: We revised this in our manuscript according to your suggestion (Line65).

Line 72 – “reactive oxygen species (ROS)-related genes” – there is no gene related to ROS

Remove lines 77-79

Response: We have deleted them in this text.

Lines 84-85 – “Trans-membrane domain prediction showed that these eight MYBs do not have a trans-membrane domain” - transmembrane not trans-membrane; this sentence should be improved;

Response: We have revised it in our manuscript according to your suggestion (Line81).

Line 166 – “The control GFP was distributed” There is no “control GFP”

Response: We have revised it in our manuscript according to your suggestion (Line173).

Line 175-176 – “Plants co-overexpressing BplMYB46 and BplMYB13 display alleviated oxidative and cell membrane damage” change to “In plants overexpressing BplMYB46 and BplMYB13 oxidative stress was alleviated and the cell membrane damage was diminished”

Response: We have revised it in our manuscript with your suggestion (Line205).

Line 194 – “plants had enhanced abilities to scavenge oxygen” change to “scavenge ROS”

Response: We have revised it in our manuscript according to your suggestion (Line226).

Lines 273-274 – “Baits and preys were transformed into Y2H cells” give the correct yeast strain name

Change “vector pbi121” to vector pBI121

Response: We revised them in our manuscript according to your suggestion (Line349-351).

Reviewer 2 Report

Major concerns:

Throughout the manuscript the authors fail to mention which statistical tests were used and what was the significance cutoff used. None of the "significantly different" statements are valid if these two issues are not addressed.

In the microscopy images, authors mention they observed differences in intensity in multiple replicates. However, they only show a representative figure which does not present how much variation was seen within each combination of genotype and condition. The authors should use an image analysis software (eg., ImageJ) to quantify the level of staining in the leaves in each condition for each replicate. They can then use a simple statistical test to determine if the level of staining is significantly different. Based on the representative image show in Fig 6 , the difference should be very significant. Please provide these details either as a figure or a supplementary table.

The authors fail to mention where they obtained the transcriptome data. Either provide a direct link to the data or provide an accession number for the database where the data is deposited.

 Minor concerns:

The manuscript fails to italicize Latin names for plant species. Please do so.

The results section has a placeholder line at its start. This should be removed.

On page 6, line 175 should be the heading of a new section.

Author Response

Throughout the manuscript the authors fail to mention which statistical tests were used and what was the significance cutoff used. None of the "significantly different" statements are valid if these two issues are not addressed.

Response: We have added the statistical tests according to your suggestion (Line407-408).

In the microscopy images, authors mention they observed differences in intensity in multiple replicates. However, they only show a representative figure which does not present how much variation was seen within each combination of genotype and condition. The authors should use an image analysis software (eg., ImageJ) to quantify the level of staining in the leaves in each condition for each replicate. They can then use a simple statistical test to determine if the level of staining is significantly different. Based on the representative image show in Fig 6, the difference should be very significant. Please provide these details either as a figure or a supplementary table.

Response: We downloaded the software of ImageJ, but found that it could not quantify the level of staining using NBT, DAB, and Evans blue, which do not emit light. We reviewed the references and did not find a study that used ImageJ to quantify the staining using NBT, DAB and Evans blue.

In our study, the MDA (levels of O2) and H2O2 contents were detected. NBT and DAB staining revealed that the cellular levels of O2 and H2O2, and the MDA (levels of O2) and H2O2 contents can verify the results of NBT and DAB staining. Evans blue staining can reflect the cell death in plants, so we added the electrolyte leakage results for of transgenic and wild-type, which can reflect cell death in plants, in Figure 7.

The authors fail to mention where they obtained the transcriptome data. Either provide a direct link to the data or provide an accession number for the database where the data is deposited.

Response: We uploaded the sequence information of the genes to NCBI and obtained the GenBank accession numbers, and added them to our manuscript (Line78).

Minor concerns:

The manuscript fails to italicize Latin names for plant species. Please do so.

Response: We revised them in our manuscript according to your suggestion.

The results section has a placeholder line at its start. This should be removed. On page 6, line 175 should be the heading of a new section.

Response: Sorry, these errors may have occurred during the editing and layout process. We have revised them.

Reviewer 3 Report

The manuscript ‘BlpMYB46 from Betula platyphylla Can Form Homodimers and Heterodimers and is Involved in Salt and Osmotic Stresses’ conveys a persuasive argument that the MYB46 protein interacts with five other MYB proteins to increase the binding efficiency to the MYBCORE to mediate ROS scavenging in response to salt and osmotic stress.  The authors used a number of unique approaches to characterize gene and protein expression and activity to support their point.  To link gene expression and interactions with the ROS scavenging hypothesis, authors recorded cell death using nitroblue tetrazolium and 3,3’-diaminobenzidine with Evans blue staining within detached leaves to visualize the effect of BpMYB46 co-expressed with BlpMYB13 compared to MYB46 alone and WT leaves.  They also measured superoxide dismutase (SOD), peroxidase (POD), and glutathione-S-transferase (GST) gene expression.  While this manuscript incorporates a number of molecular approaches it lacks sufficient explanation of the results and logic behind the experiment to fully support the conclusions.  It’s my opinion the majority of these gaps could be rectified by simple explanatory sentences explaining the authors logic and/or reasoning.  

Genus species names should be italicized throughout the text and references

Throughout the text gene names should be italicized, while protein names should remain as plain text. This will vastly improve the readability of the manuscript. 

Line77-79, remove the instructions. 

Figure 1:  BlpMYB46 constructs interact with everything in the DDO media, though there is no negative control represented.  The important point is the QDO interactions.  This information is presented in the paragraph above the figure, but is not explained in the figure legend.

Secion 2.5.  Why was mannitol used as the osmotic stresser?  I don’t think this is explained anywhere in the paper and for researchers not familiar with mannitol, this is a major gap. 

When looking at figure 4, the different MYBs have different responses.  However, reading the text of the manuscript it seems they’re all similar. Please highlight and hypothesize on the differences.  For example, the six MYBs appear to split into 2 response groups for NaCl stress (group 1: MYB11, 6, 8 and group 2: Myb46, 12, and 13), but three response groups under Mannitol stress (MYB13 and 46 in group 1, MYB6, 8, and 12 in group 2, and MYB 11 in group 3).  Given these differences, I’d hypothesize unique responses for each group to the different stresses.

Line 209.  Please spell out SOD, POD, and GST here.  Yes, they are in the M&M, but this is their first appearance. 

Figure 2.  Because the spacing between each of the bars is so wide, it’s difficult to tell which columns are statistically unique or, conversely, which columns statistically overlap.  Please add letters (A, B, C) to the right of each bar to indicate statistical significance.  

Figure 6.  This is a nice figure, but without BlpMYB13 alone represented, it’s not clear to me that the co-expression is important, rather it could just be that BlpMYB13 induces a greater response.  

Why were follow-up experiments only performed with BlpMYB13.  I understand it showed the most significant difference from MYB46 based on Figures 2 and 3, but are all MYB46 interactions identified in this experiment expected to be similar? Why or why not?

Author Response

Genus species names should be italicized throughout the text and references. Throughout the text gene names should be italicized, while protein names should remain as plain text. This will vastly improve the readability of the manuscript. 

 Response: We revised them in our manuscript according to your suggestion.

Line77-79, remove the instructions. 

Response: Sorry, these errors may have occurred during the editing and layout process. We have deleted them.

Figure 1:  BlpMYB46 constructs interact with everything in the DDO media, though there is no negative control represented.  The important point is the QDO interactions.  This information is presented in the paragraph above the figure, but is not explained in the figure legend.

Response: We added have added the information that the bait pGBKT7-53 and the prey pGADT7-Rec-T were used as a positive control (Line329-333). When the bait and pray were co-transformed into yeast cells, they can grow in DDO medium, indicating that the bait protein and pray protein were expressed, as long as the yeast transformation experiment is successful. There were negative controls in the QDO interactions, and we revised the figure legend 1 accordingly.

Secion 2.5.  Why was mannitol used as the osmotic stresser?  I don’t think this is explained anywhere in the paper and for researchers not familiar with mannitol, this is a major gap. 

Response: Mannitol is a naturally occurring sugar alcohol that can cause osmotic stress and has been widely used for studies of osmotic stress in plants (Kumari, A.; Jewaria, P. K.; Bergmann, D. C.; Kakimoto, T., Arabidopsis reduces growth under osmotic stress by decreasing SPEECHLESS protein. Plant & cell physiology 2014, 55, (12), 2037-2046). We have added the explanation in section 3.8 of the Materials and Methods (Line371-372).

When looking at figure 4, the different MYBs have different responses.  However, reading the text of the manuscript it seems they’re all similar. Please highlight and hypothesize on the differences.  For example, the six MYBs appear to split into 2 response groups for NaCl stress (group 1: MYB11, 6, 8 and group 2: Myb46, 12, and 13), but three response groups under Mannitol stress (MYB13 and 46 in group 1, MYB6, 8, and 12 in group 2, and MYB 11 in group 3).  Given these differences, I’d hypothesize unique responses for each group to the different stresses.

Response: Thank you for your suggestions. We have added the detailed descriptions about the expression analyses of BplMYB46 and the other five BplMYBs under salt and osmotic stress (Line184-196).

Line 209.  Please spell out SOD, POD, and GST here.  Yes, they are in the M&M, but this is their first appearance. 

Response: Thank you for your suggestion. We have added the full names of these genes at this point in the manuscript (Line243-244).

Figure 2.  Because the spacing between each of the bars is so wide, it’s difficult to tell which columns are statistically unique or, conversely, which columns statistically overlap.  Please add letters (A, B, C) to the right of each bar to indicate statistical significance.  

Response: We have revised the figure according to your suggestion (Line145-148, Line407).

Figure 6.  This is a nice figure, but without BlpMYB13 alone represented, it’s not clear to me that the co-expression is important, rather it could just be that BlpMYB13 induces a greater response.  

Response: We have added the analyses of BplMYB13 overexpression in our manuscript (in sections 2.7, 2.8 and 2.9 of the Results and Discussion).

Why were follow-up experiments only performed with BlpMYB13.  I understand it showed the most significant difference from MYB46 based on Figures 2 and 3, but are all MYB46 interactions identified in this experiment expected to be similar? Why or why not?

Response: We further analyzed the expression patterns of BplMYB46 and BplMYB6, 8, 11, 12, and 13 (Line184-196), in addition to the interaction of BplMYB46 with the five MYB proteins. The results indicated that BplMYB46 and BplMYB13 shared very similar expression patterns under salt or osmotic stress; therefore, BplMYB13 was selected for further study under stress (Line207-209).

Round 2

Reviewer 1 Report

Line 15: „Overexpression of BplMYB46 from Betula platyphylla” - add the species of over-expressing plant; if it was not birch change “overexpression” to “ectopic expression”.

Line 16: “the interaction of eight MYB transcription factors” change to “the interaction of eight MYB transcription factors from Betula platyphylla”

Line “ Real-time fluorescence quantitative PCR” change to „ Real-time quantitative PCR”

Lines 37-38: “MYB113 or 37 MYB114” change to “AtMYB113 or AtMYB114

Line 39: “MYBH” change to “AtMYBH

Lines 45-46 ”MYB proteins share a conserved MYB DNA-binding domain that binds to cis-acting elements of transcription factor genes” – does MYB binds only to TF genes?

Lines 50-51: “B-MYB forms a complex with itself to influence its transcriptional activity” specify species

Line 56: “MYB75” change to “AtMYB75

Line 58: “MYB5 and MYB14” change to “MtMYB5 and MtMYB14”

Lines 61-62: change “the interactions among MYB proteins or between MYBs and other proteins” change to “the interactions among MYB proteins or between MYBs and transcription factors from the other families”

Remove all traces of tracking changes (letters in red)